# Polyphenol and flavonoid content in major Teff [*Eragrostis tef (Zuccagni) Trotter*] varieties in Amhara Region, Ethiopia

**Chaltu Reta**[1], **Minaleshewa Atlabachew**[1]*, **Tihitinna Asmellash**[1], **Woldegiorgis Hilluf**[1], **Marie Yayinie**[1,2], **Tessera Alemneh Wubieneh**[3]

**1** Department of Chemistry, College of Science, Bahir Dar University, Bahir Dar, Ethiopia, **2** Department of Chemistry, College of Natural Science, Debre Tabor University, Debre Tabor, Ethiopia, **3** Material Science and Engineering Department, College of Science, Bahir Dar University, Bahir Dar, Ethiopia

* atminale2004@yahoo.com

## Abstract

Teff [*Eragrostis tef* (Zuccagni) Trotter] is a small-sized cereal grain and an indigenous crop in Ethiopia. The Amhara region is one of the major teff producers regions in the country. However, information on the phenolic content of the region's teff varieties is limited. Seventy-two teff samples were collected from three administrative zones (West Gojjam zone, Awi zone, and East Gojjam zone) of the Amhara region of Ethiopia. The samples' total polyphenol and flavonoid contents were determined using colorimetric methods. The total flavonoid contents expressed as catechin equivalent, CE (i.e., under alkaline conditions) and quercetin equivalent, Q.E (i.e., under the methanolic solution of $AlCl_3$) were found to be in the range of 7.66 ± 0.60–57.36 ± 3.87 mg C.E and 15.45 ± 0.15–113.12 ± 3.09 mg Q.E per 100 g of teff samples, respectively. The corresponding total polyphenol content (TPC), described as gallic acid equivalent (G.A.E.), was in the range of 46.21 ± 1.20–133.32 ± 5.44 mg G.A.E. The results showed that the mean TPC value of the teff samples from the West Gojjam zone was enriched with polyphenol than samples from the Awi zone and East Gojjam. Furthermore, it was noted that the mean TPC and TFC values did not vary significantly between samples of the East Gojjam and Awi zone (p > 0.05). In contrast, a significant difference in mean TPC and TFC-Q.E were noted between the sampling zone of East Gojjam and West Gojjam and between West Gojjam and Awi zones (p < 0.05). These significant variations in TPC and TFC might be due to observable variations in the agroecological zones and the genetic—make-up of the samples. Person correlation indicated a significant positive correlation matrix between the three variables (p = 0.01). The teff samples were trying to be classified based on their geographical origin using hierarchical cluster analysis (HCA) and biplots. Accordingly, the variance explained by component 1 (PC1) is 67.2%, and the variance explained by component 2 (PC2) is 20.0%.

**Data Availability Statement:** All the data are included in the paper.

**Funding:** Bahir Dar University, college of science, has provided laboratory facilities, covered the

tuition fee of the PhD student, Chaltu Reta and provided her with study leave. The research and community service vice president office of Bahir Dar University has supported costs related to sample collection. The funders had no role in study design, data collection and analysis, decision to publish, or preparation of the manuscript.

**Competing interests:** The authors have declared that no competing interests exist

# Introduction

Teff [*Eragrostis tef* (Zuccagni) Trotter] is an ancient small-sized tropical cereal that has its center of origin and has diversified in the Ethiopian highlands, where it is believed to have been domesticated [1]. It is used as a cereal crop in the *Poaceae* or *Gramineae* family with small grains and originated in Ethiopia between 4000 and 1000 BC [2]. In other countries, like Australia, South Africa, and the United States, it is used as a forage crop for animal feed and as a thickener for soups, stews, and gravies [3].

Teff has a naturally higher nutritional value and is rich in essential amino acids and minerals such as Fe, Zn, Mg, and Ca than cereals such as maize and pearl millet. On the other hand, a comparable level of protein and carbohydrates is reported in teff grains compared to the most common cereal grains such as wheat, barley, sorghum, and pearl millet, and it is a gluten-free cereal grain [4,5]. As many people are diagnosed with celiac disease and other forms of gluten sensitivity, the demand for gluten-free meals is increasing. As a result, the nutrient profile of teff grain suggests that it has a lot of potential for application in foods and beverages around the world [2,6].

Teff is a staple food in Ethiopia, and the flour of teff is mainly used for making "injera" which is the favourite national dish. The "injera" prepared from teff has a pleasing odour, flavor, texture, and quality [7,8]. Teff is a chasmogamous C4 annual cereal with a fibrous root system and usually upright stems, while some cultivars have to bend or elbow varieties [9,10].

Teff grain is tiny and occurs in various colors, ranging from pale white to ivory white, light brown to dark brown, and reddish-brown purple. This color determines grade and market price [10,11].

Teff is adapted to growing environmental conditions such as drought and waterlogging. It can thrive in some harsh climate conditions when major crops might fail [7]. It is also less vulnerable to weevils and other pests during storage than other cereal grains [5]. It has the greatest area under cereal cultivation and is Ethiopia's third-largest grain producer (after maize and wheat) [12,13].

Different teff varieties are on the market. In terms of color, the most common teff varieties are white, brown, and red. Brown teff is also known as mixed-colored teff. It contains natural mixed white and red-colored teff grains. The report indicated significant variation in minerals and other nutritional constituents between the three colored variants [14,15]. The white variety of teff is smaller in size. Red teff has better mineral and essential amino acid compositions and slightly higher carbohydrate, fibre, and energy content [4,16]. White teff is the most widely used teff among customers and is mainly grown in Ethiopia's highlands, under the most challenging conditions. It is the most expensive type of teff [7].

Whole-grain cereal intake has been provided to effectively reduce the risk of cardiovascular disease, type 2 diabetes, ischemic stroke, obesity, and cancer [17,18]. Phenolic, Fiber, and other bioactive components are the quickest to attribute to these therapeutic benefits [19–21].

Polyphenols are secondary plant metabolites that adequately defend plants from diseases and ultraviolet radiation [22]. In addition, polyphenolic compounds also protect cell constituents from oxidative damage, lowering the risk of oxidative stress-related illnesses [23,24].

Polyphenols such as flavonoids and phenolic acids are found in cereal grains in different proportions. Their concentrations are significantly affected by the genetic make-up of the crop and environmental conditions. These phytochemicals are responsible for the consumer's health benefits [25,26].

As information obtained from Amhara Region's Bureau of Agriculture, about 42 improved varieties of teff have been registered in the country, but only a few are widely cultivated in the region. Among the varieties of teff, Kuncho teff (DZ-CR-387) and Tseday teff

(DZ-CR-37) are white by color and widely cultivated in the northwestern part of the Amhara region. Kuncho teff is cultivated in East Gojjam and West Gojjam zones, while Tseday teff is mainly grown in the Awi zone. In addition, red teff and brown teff varieties are also cultivated in the region, but consumers prefer these colored varieties less. Brown (mixed) teff is a variety of teff that is locally called "sergegna" and naturally exists as a mixture of white and red-colored teff varieties. Kuncho and Tseday teff varieties have a high market price and are most preferred by consumers. But the same variety of teff produced in different zones and districts have different qualities and market prices, hence having consumer preferences one over the other. Since the Amhara region is one of the most teff-producing areas of the country, there is no report on the level of total phenolics and flavonoids in different teff varieties cultivated in this region.

Thus, this study was intended to quantify the level of total phenolic and total flavonoid contents in white teff varieties (Kuncho teff (DZ-CR-387) and Tseday teff (DZ-CR-37)), Brown (Mixed) teff, and, Red teff varieties collected from 21 sub-districts of three zones (East Gojjam, West Gojjam, and Awi). Furthermore, the study aimed to see the variation of total flavonoid content with the two procedures and perform some classification models based on their geographical origins using the total phenolic and flavonoid contents.

## Materials and methods

### Sample collection and Pre-treatment

Seventy-two (72) teff samples were directly collected from model farmers in the three administrative zones of the Northwestern part of the Amhara region of Ethiopia. Based on the information obtained from the Amhara region's agriculture bureau, the Northwestern Amhara region is one of the country's most significant teff-producing areas.

Four varieties of teff such as Kuncho (white teff), Tseday (white teff), Brown (mixed), and Red teff were collected from model farmers in January 2020. About 500 g of each teff variety sample was collected and stored in an airtight plastic bag. From the East Gojjam zone, three districts (Aneded, Shebelberenta, and, Enemay) and nine sub-districts (Jamadidik, Gudalema, Adisge yegewera, Yeidwuha town, Weregona akababiw, Gedaiyesus,Weyira gurazam, Huleta-miba dibisa, and Enekorna adis amba) were selected. From the West Gojjam zone, two districts (Goinjkolela and Adet/Yilmanadensa) and seven sub-districts (Zanat, Goinj, Kore tenkere, Mentadeber, Kilelet, Adet zuria, and Mosbo) were selected. Five sub-districts (Gisayita, Gangana, Ahiti, Zigem town, and, Gudarjawwi) were chosen from the Zigem district of the Awi zone. Fig 1 shows the map of sampling sub-districts and zones. After samples were transferred to the laboratory, any contamination like dust particles, soil, and husk were separated by sieving with a 0.5mm mesh size sieve and milled with an electronic grinder.

### Chemicals, reagents, and instruments

Folin-Ciocalteu reagent, gallic acid, Catechin, Quercetin and Methanol (from Sigma–Aldrich Chemie, Steinheim, Germany), $AlCl_3$ (from Fluka Goldie, Mumbai, India), $Na_2CO_3$ and NaOH (from BLULUX Laboratories, India), and $NaNO_2$ (from Guangzhou Jinhuada Chemical reagent Co., Ltd, China) were obtained. UV-VIS spectrophotometer (Cary60, Agilent Technologies, USA) was used to measure the prepared standards and sample solutions absorbance. A quartz cuvette with a 1 cm path length was used as a sample holder., DAIHAN Scientific ultrasonic cleaner (WUC-D10H, Korea) was used for sample extraction, a Centrifuge (800–1, Japan) was used to centrifuge the extracts, and deionized water was used throughout the experiment.

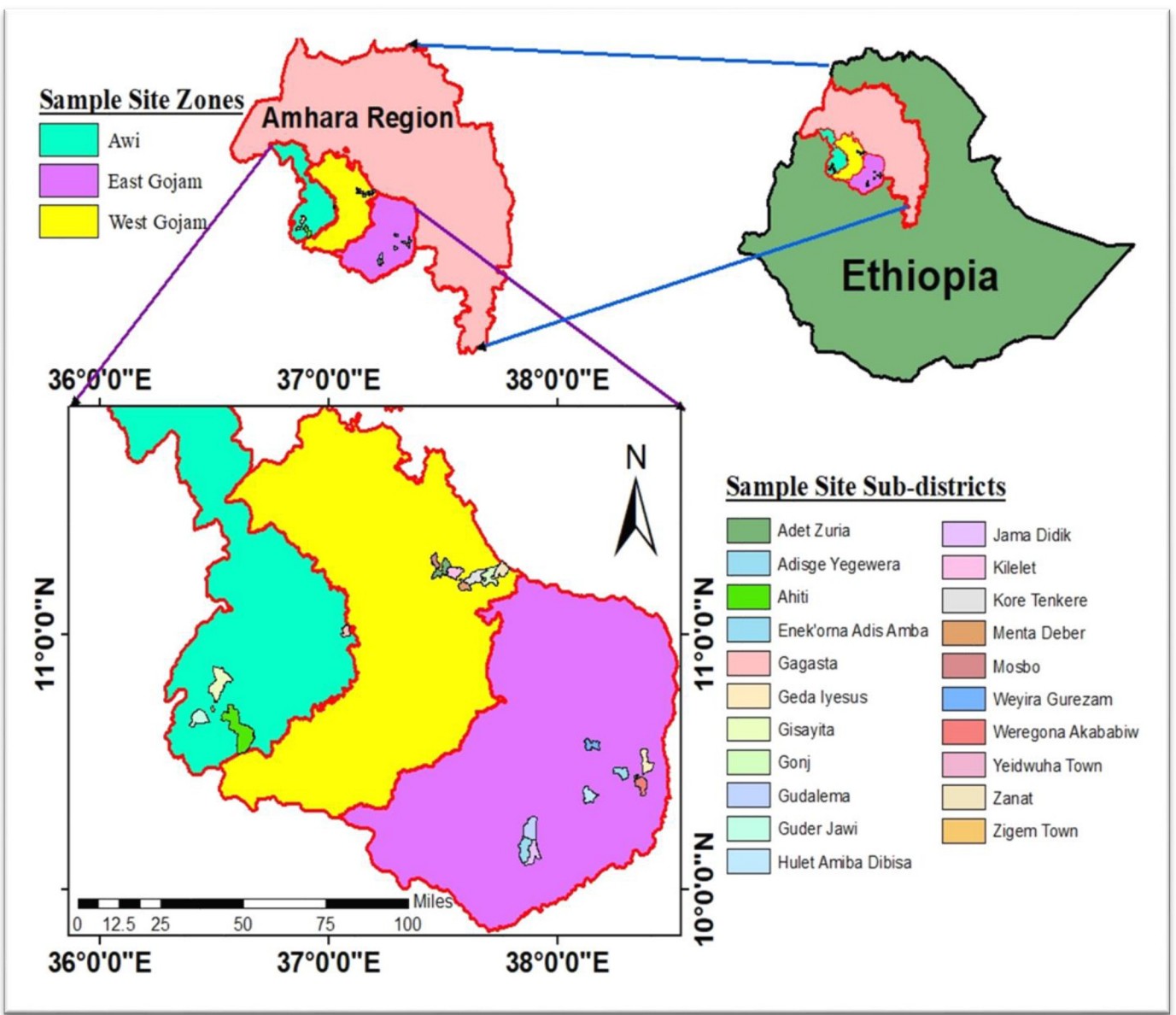

**Fig 1. Map of Amhara regional state administrative zones and sub-districts showing the sampling sites of teff samples.**

## Optimization of the extraction method

Teff flour (1.0 g) was taken in a 50 mL nylon centrifuge tube and soaked for 2 hours in 7 mL of 80:20 methanol-distilled water. The wetted sample was extracted under an ultrasonic water bath [27] for 30 min, 45 min, and 60 min at a fixed temperature (35°C). The mixture was centrifuged for 15 min at 3500 rpm, where the supernatant was decanted carefully into a second centrifuge tube. The residue was re-extracted with the same solvent ratio and volume for the second time. The supernatants from the two extracts were combined and filtered through a nylon membrane syringe filter with a 0.45μm pore filter. It was noted that 45 min extraction time provided higher total phenolic content and was used as an optimum extraction time for all the samples.

## Determination of total phenolic content by Folin–Ciocalteu assay

The Folin–Ciocalteu assay was performed to estimate the total phenolic content (TPC) in the extracts following the procedure developed by Singleton et al. [28] and modified by Atlaba-chew et al. [29]. Briefly, 0.6 mL of the extract was mixed with 1 mL of Folin–Ciocalteu reagent and 4 mL of distilled water. After 5 min, 4 mL of 7.5% sodium carbonate solution was added and kept in the dark for 2 hours at ambient temperature before taking the absorbance at 765 nm. Similarly, a calibration curve was constructed using gallic acid standard over a concentration range of 1 mg/L to 100 mg/L and reported as mg gallic acid equivalent per 100 g of teff sample (mg G.A.E./100 g). Each measurement was conducted in triplicate.

## Determination of total flavonoid content as catechin equivalent (TFC-C.E)

In this study, the catechin-like flavonoids were estimated as catechin equivalent following the method reported by Pekal and Pyrzynska [27] using $AlCl_3$, $NaNO_2$, and NaOH solutions. Briefly, 2 mL of each catechin standard solution or sample solution was taken and treated with 0.6mL of 5%$NaNO_2$. After 6 min, 0.6 mL of 10% $AlCl_3$ solution was added, shaken, and left to stand for 6 min. Finally, 4mL of 4% NaOH solution was diluted to 8 mL with 80% aqueous methanol. After 15 min, absorbance was taken at 510 nm using a UV-VIS spectrophotometer [30]. The calibration curve was constructed over a concentration range of 1 mg/L to 50 mg/L of catechin, and results were expressed as mg catechin equivalent per 100 g of sample (mg C. E./100g). Each sample was analysed in triplicate.

## Determination of total flavonoid content as quercetin equivalent (TFC- Q.E)

Quercetin-like flavonoids were determined using a methanolic solution of aluminum chloride. Exactly 1mL of sample solution was taken and treated with 5 mL of 2% $AlCl_3$ dissolved in methanol and was diluted to 10 mL with distilled water and left to stand for 60 min until the yellow color was developed. Finally, the absorbance of the resulting solution was measured at 425 nm [31,32]. The calibration curve was constructed over 0.1–80 mg/L of quercetin, and results were expressed as mg quercetin equivalent per 100 g of sample (mg Q.E./100g). Each sample was analyzed in triplicate.

## Statistical data analysis

One-way variance (ANOVA) was used to evaluate the mean values of total phenolic content (TPC), total flavonoid content as quercetin equivalent (TFC-Q.E), and total flavonoid content as catechin equivalent (TFC-C.E) at 95% confidence level using Stata .14, Origin 8.0, IBM SPSS statistics 23, and Excel statistical soft-wares. The ANOVA analysis was used to know whether the variation of the mean of polyphenol content and flavonoid content was significant or not between the sampling zones. The observed datasets were chemometrically handled using Principal component analysis (PCA) to see the loading eigenvectors and the extent of variations. Principal Component Analysis (PCA) is a technique used for data reduction and improves the interpretability of data with low information loss.

## Results and discussion

The UV-Vis absorbance versus wavelength spectra and calibration curves of gallic acid, catechin, and quercetin standards used are depicted in Figs 2A and 2B, 3A and 3B and 4A and 4B, respectively. A good linear response over a concentration ratio of 1.00–100.00 mg/L, 1.00–50.00 mg/L, and 0.10–80.00 mg/L, respectively, for gallic acid, catechin, and quercetin, respectively, was noted.

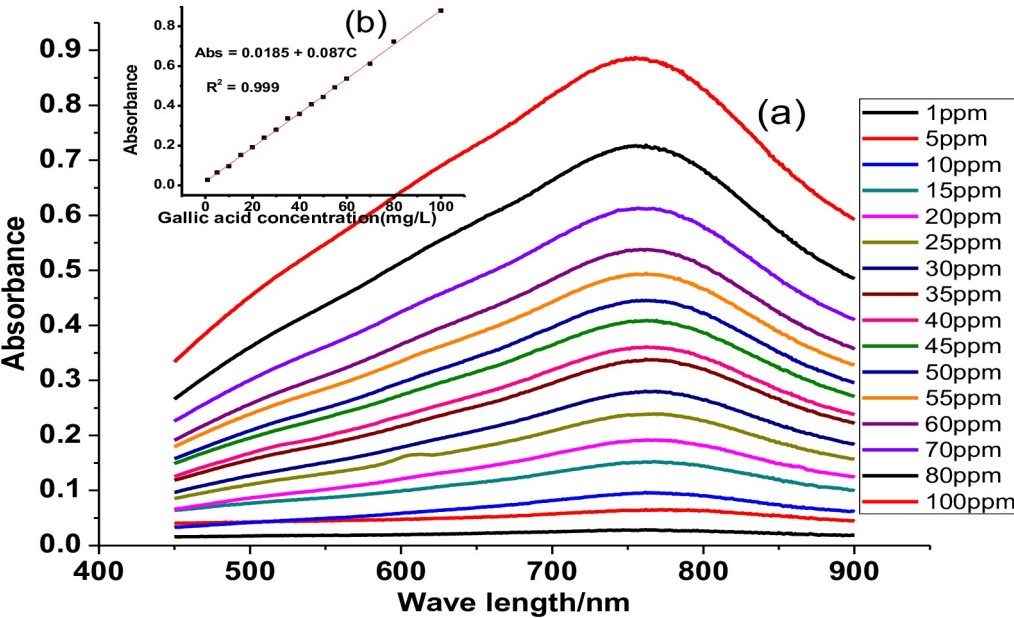

**Fig 2.** UV-Vis spectra of Gallic acid standard (a) and its calibration curve (b).

The obtained results for 72 teff samples collected from the model farmers of different geographical origins are reported in Table 1.

The highest TPC (133.32 mg G.A.E./100g) was noted in the sample from the Gisayita sub-district of the Awi zone. At the same time, the minimum concentration (46.21 mg G.A.E./100 g) was obtained in the sample from the Gedaiyesus sub-district of the East Gojjam zone. As indicated in Table 1, the total phenolic content in teff samples from the Awi zone ranged between 53.77–133.32 mg G.A.E./100g of teff samples. On the other hand, the phenolic

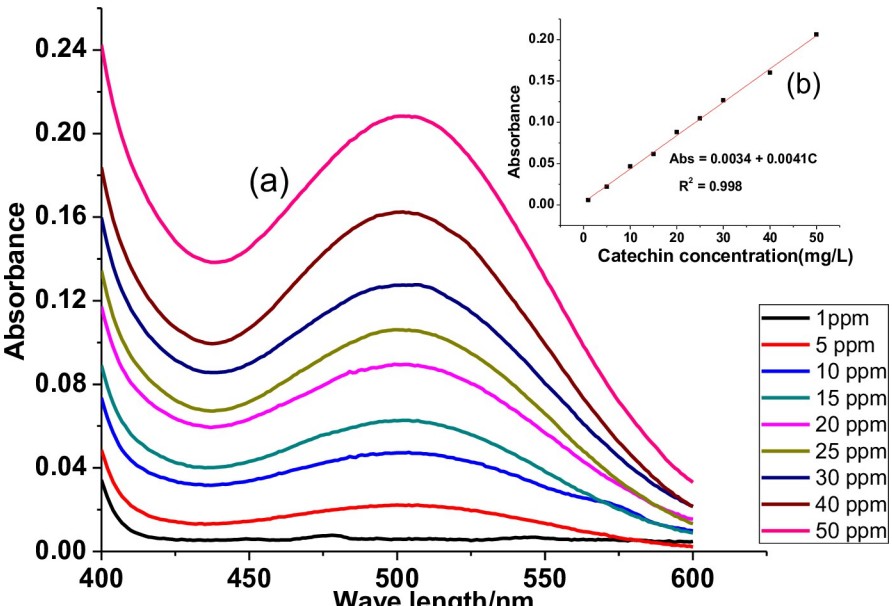

**Fig 3.** UV-Vis spectra of Catechin standard (a) and its calibration curve (b).

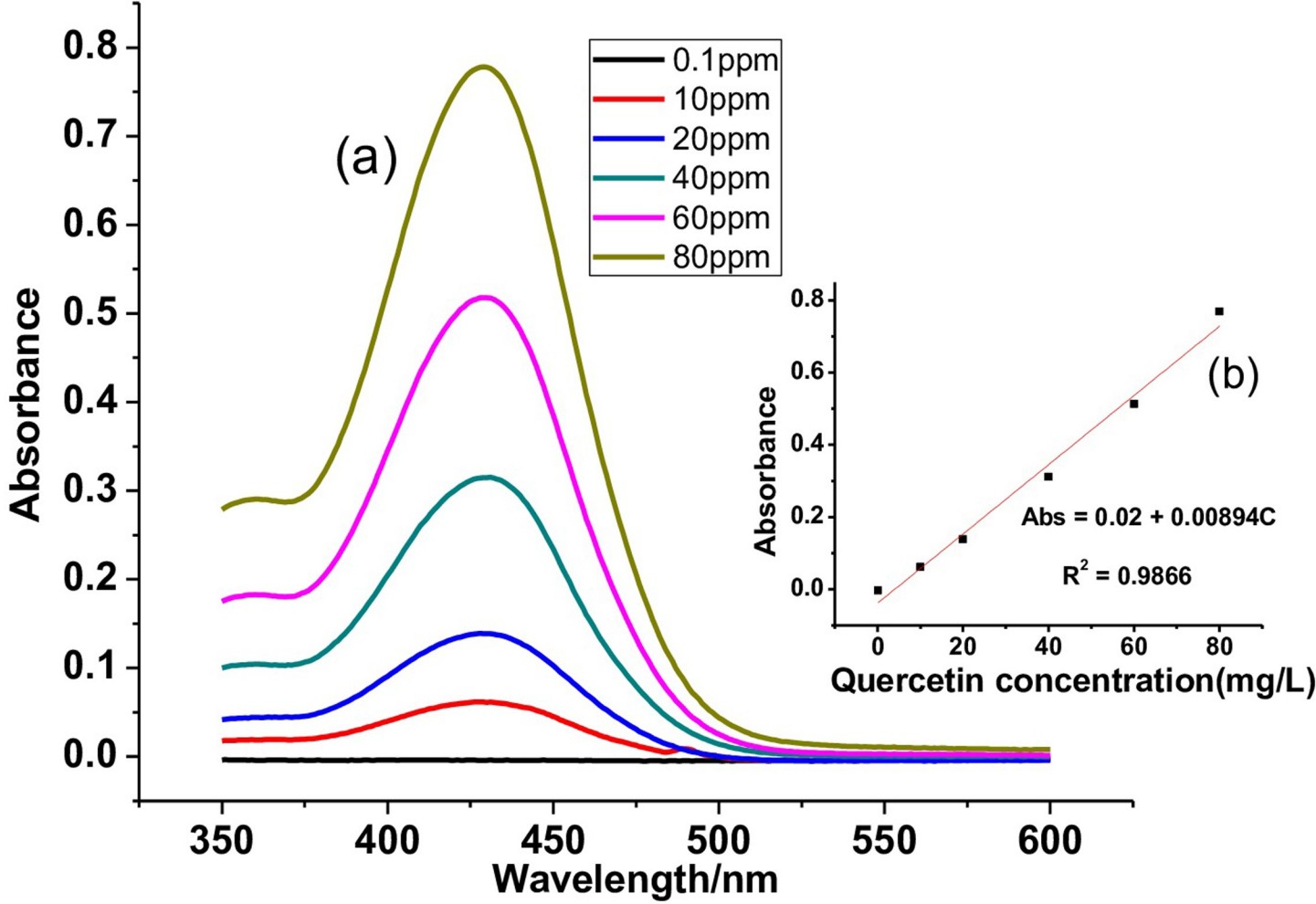

**Fig 4.** UV-Vis spectra of Quercetin standard (a) and its calibration curve (b).

content recorded in samples from the East Gojjam and West Gojjam zones ranged between 46.21–92.20 mg G.A.E/100g and 46.60–128.51 mg G.A.E./100g, respectively.

The brown or mixed teff variety from Awi zone contained a higher total phenolic content than the white teff (Tseday variety) collected from the Awi zone. The mean TPC values were compared between the two white teff varieties (Kuncho and Tseday) collected from the three administrative zones. It was noted that the Kuncho teff variety collected from the West Gojjam Zone contained a higher mean TPC value than the same Kuncho variety collected from the East Gojjam zone and the other variety (Tsedey) from the Awi zone. The maximum TPC was about 2.85 times as high as its minimum TPC value for each sampling zone.This indicates high variability in TPC among samples collected from different sampling zones.

The TPC results of the present study were compared with reported data. An unknown variety of samples collected from the Netherlands and Italy were found to have a TPC value of 175.65 mg G.A.E./100 g and 173.20 mg GAE/100g, respectively [33,34]. Similarly, brown teff from Bolivia and USA contained 186 mg G.A.E./100g and 219 mg G.A.E./100g, respectively. Whereas 142 mg G.A.E./100g and 141 mg G.A.E./100g of TPC were noted in white teff samples from these two countries with the same order [25]. An unknown variety of teff collected

**Table 1. Concentration (mean±standard deviation) of TPC (mg G.A.E./100g), TFC-C.E (mg C.E./100g), and TFC-Q.E (mg Q.E./100 g) of teff samples from 21 sub-districts of Amhara region.**

| zones | Districts | Sub-districts | Types of Teff | Concentration of TPC in mg G.A.E./100g | Concentration of flavonoid in mg C.E. /100g | Concentration of flavonoid in mg Q.E. /100g |
|---|---|---|---|---|---|---|
| East Gojjam | Aneded | Jamadidik | Kuncho | 70.17±2.42 | 10.36±0.80 | 30.29±0.30 |
| | | | Kuncho | 78.92±6.10 | 19.06±0.43 | 36.86±0.77 |
| | | | Kuncho | 53.34±2.78 | 9.89±0.11 | 28.91±1.16 |
| | | Gudalema | Kuncho | 70.30±4.32 | 20.01±0.73 | 30.83±1.78 |
| | | | Kuncho | 62.24±0.45 | 16.47±0.73 | 25.37±1.82 |
| | | | Kuncho | 64.21±2.12 | 47.91±4.24 | 58.74±4.12 |
| | | Adisge yegewera | Kuncho | 69.43±3.17 | 10.73±0.59 | 24.59±1.23 |
| | | | Kuncho | 78.50±1.62 | 11.78±0.05 | 31.95±2.02 |
| | | | Kuncho | 70.90±0.43 | 15.83±0.73 | 31.38±0.48 |
| | Shebel berenta | Gedaiyesus | Kuncho | 61.30±1.99 | 13.12±0.65 | 25.95±1.17 |
| | | | Kuncho | 85.57±4.81 | 22.73±0.15 | 29.83±0.21 |
| | | | Kuncho | 78.49±4.66 | 13.73±0.37 | 27.22±0.89 |
| | | | Kuncho | 46.21±1.54 | 7.66±0.60 | 15.45±0.15 |
| | | | Kuncho | 87.61±5.05 | 14.97±0.14 | 23.77±1.05 |
| | | | Kuncho | 82.55±6.57 | 15.87±0.33 | 24.94±1.88 |
| | | Yeidwuha town | Kuncho | 64.66±0.68 | 16.80±0.09 | 30.81±0.32 |
| | | | Kuncho | 58.06±3.65 | 16.36±1.17 | 23.88±1.54 |
| | | | Kuncho | 92.20±3.06 | 15.31±0.45 | 22.62±1.39 |
| | | Weregona akababiw | Kuncho | 75.15±2.48 | 54.30±0.73 | 60.53±5.17 |
| | | | Kuncho | 64.49±4.68 | 16.09±0.78 | 29.96±0.42 |
| | | | Kuncho | 66.73±2.40 | 14.11±0.45 | 20.60±1.35 |
| | Enamey | Weyira gurazem | Kuncho | 79.51±4.47 | 32.57±0.89 | 27.38±0.02 |
| | | | Kuncho | 68.00±1.46 | 22.96±0.81 | 29.62±1.46 |
| | | | Kuncho | 60.12±3.00 | 17.86±0.35 | 27.16±1.47 |
| | | Huletamiba dibisa | Kuncho | 70.38±0.42 | 22.27±0.28 | 29.64±1.35 |
| | | | Kuncho | 77.86±2.57 | 26.37±1.40 | 29.56±1.52 |
| | | | Kuncho | 72.33±3.07 | 24.66±0.04 | 29.48±1.05 |
| | | Enekorna Adisamba | Kuncho | 73.81±0.01 | 35.13±1.71 | 36.57±1.21 |
| | | | Kuncho | 77.96±5.02 | 32.82±0.95 | 33.93±1.51 |
| | | | Kuncho | 79.98±1.90 | 36.62±2.04 | 33.98±1.52 |
| West Gojjam | Gonjikolela | Zanat | Kuncho | 77.14±1.09 | 17.10±0.33 | 36.64±2.66 |
| | | | Kuncho | 77.11±2.55 | 19.12±0.77 | 31.23±2.60 |
| | | | Kuncho | 86.34±2.84 | 17.22±0.38 | 46.38±0.60 |
| | | Gonj | Kuncho | 122.63±2.68 | 26.34±0.33 | 50.43±1.14 |
| | | | Kuncho | 118.52±1.31 | 35.72±1.58 | 33.08±1.32 |
| | | | Kuncho | 66.87±1.77 | 19.02±0.71 | 42.74±1.20 |
| | | Kore tenkere | Kuncho | 51.07±2.26 | 17.34±0.19 | 20.83±1.68 |
| | | | Kuncho | 116.89±3.03 | 24.42±0.18 | 60.00±4.28 |
| | | | Kuncho | 113.29±8.86 | 31.62±0.82 | 69.52±2.05 |
| | | Mentadeber | Kuncho | 123.00±8.24 | 28.35±0.05 | 81.23±3.77 |
| | | | Kuncho | 98.56±5.27 | 19.78±0.61 | 38.39±3.41 |
| | | | Kuncho | 105.26±1.53 | 18.78±0.50 | 79.30±2.22 |
| | Adet/ Yilmanadensa | kilelet | Kuncho | 110.10±4.31 | 26.36±1.67 | 40.57±1.67 |
| | | | Kuncho | 108.91±2.67 | 19.15±0.90 | 57.98±4.34 |
| | | | Kuncho | 128.51±1.20 | 24.14±1.58 | 58.51±1.27 |

(*Continued*)

**Table 1.** (Continued)

| zones | Districts | Sub-districts | Types of Teff | Concentration of TPC in mg G.A.E./100g | Concentration of flavonoid in mg C.E. /100g | Concentration of flavonoid in mg Q.E. /100g |
|---|---|---|---|---|---|---|
| | | Adet zuria | Kuncho | 92.02±2.03 | 25.37±0.48 | 87.74±1.98 |
| | | | Kuncho | 89.66±2.05 | 30.45±1.03 | 78.52±2.05 |
| | | Mosbo | Kuncho | 46.60±1.20 | 15.94±0.93 | 37.19±2.77 |
| | | | Kuncho | 78.12±2.81 | 28.93±2.50 | 48.20±4.11 |
| | | | Kuncho | 90.95±7.86 | 22.59±0.46 | 53.28±0.40 |
| | | | Kuncho | 94.99±2.10 | 19.50±0.22 | 74.90±3.48 |
| Awi | Zigam | Gisayita | Brown (mixed) | 104.44±3.42 | 35.58±1.70 | 48.31±1.90 |
| | | | Brown (mixed) | 133.32±5.44 | 57.36±3.87 | 113.12±3.09 |
| | | | Brown (mixed) | 117.44±2.50 | 34.04±2.52 | 42.66±1.55 |
| | | | Tseday | 72.94±1.63 | 18.03±0.49 | 60.39±2.27 |
| | | | Tseday | 70.54±0.64 | 17.68±0.63 | 36.61±2.52 |
| | | | Tseday | 98.58±4.16 | 23.48±0.24 | 30.74±2.36 |
| | | | Red | 86.32±0.25 | 26.80±1.96 | 80.18±0.09 |
| | | | Red | 80.25±2.92 | 50.93±2.90 | 73.80±2.97 |
| | | | Red | 80.24±0.91 | 48.25±3.30 | 73.64±4.94 |
| | | Zigam town | Tseday | 68.52±1.14 | 18.63±1.65 | 37.49±0.86 |
| | | | Tseday | 75.56±3.21 | 17.42±0.29 | 39.66±3.02 |
| | | | Tseday | 69.80±3.62 | 19.29±0.01 | 38.32±3.65 |
| | | Ahiti | Tseday | 75.44±4.08 | 16.29±0.46 | 44.10±1.90 |
| | | | Tseday | 68.86±0.32 | 21.38±0.36 | 33.48±2.02 |
| | | | Tseday | 75.55±2.73 | 20.54±0.59 | 57.22±0.86 |
| | | Gagasta | Tseday | 88.80±3.58 | 21.50±1.78 | 37.13±0.43 |
| | | | Tseday | 58.92±1.73 | 11.33±0.58 | 35.39±1.34 |
| | | | Tseday | 53.77±2.21 | 17.78±0.68 | 30.38±1.68 |
| | | GudarJawi | Tseday | 66.29±0.68 | 15.88±0.54 | 23.05±1.05 |
| | | | Tseday | 62.22±2.68 | 18.18±0.88 | 31.52±0.73 |
| | | | Tseday | 64.01±0.52 | 20.04±0.72 | 34.18±2.51 |

from a marketplace in Addis Ababa, Ethiopia, reported a TPC of 123 mgG.A.E./100g [35]. All these reported data were relatively higher than the mean TPC value of the present study (71–95 mg G.A.E./100g of teff samples). The possible reason for such variation might be differences in genetic resources of the raw material, agronomic practice, and environmental conditions of the sampling regions. On the other hand, 263 mgG.A.E/100g and 448 mgG.A.E /100g, respectively, for white (Tseday type) and brown teff varieties collected from the Tigray region of Ethiopia were reported by [14]. In this study, the total flavonoids in the 72 teff samples were determined following two approaches. Reports indicate that not all flavonoids react with $AlCl_3$ in the presence of sodium nitrite and sodium hydroxide. Instead, flavonols and flavones classes of flavonoids such as quercetin, quercetrin, galangin, and rutin, are selectively determined at 410–430 nm by reacting the sample solutions with $AlCl_3$ in acidic solution acetate buffer or methanolic solutions. In contrast, flavan-3-ols (catechins and others) could selectively react with $AlCl_3$ solution in sodium nitrite and alkaline conditions, and measurement can be done at 510 nm [30]. Thus, flavonoid data obtained with only one of the approaches could not reflect the total flavonoid content in a given sample. The total flavonoid content as quercetin

equivalent (TFC-Q.E) ranged from 15.45–113.12 mg Q.E./100g of teff sample, as indicated in Table 1. The lowest and highest TFC-Q.E were found in samples from the East Gojjam zone (15.45 mg Q.E./100g) and the Awi zone (113 mg Q.E./100g). Generally, teff samples from the Awi zone had the highest TFC compared to the other two zones. This comparative difference might be due to the varietal type effect and other environmental conditions. The total flavonoid content expressed as catechin equivalent ranged from 7.66–57.36 mg C.E./100g. The lowest concentration of TFC-C.E was noted in sample from the East Gojjam zone, and the highest was recorded in the teff sample analyzed from the Awi zone.

In this study, the amount of flavonoid as quercetin equivalent was higher than the total flavonoid content as determined with catechin equivalent. On average, the TFC-Q.E content in the teff samples is 1.5 times higher than the TFC done with catechin equivalents. Similar to the TPC value, the brown (mixed) teff samples from the Awi zone had higher flavonoid contents (Table 1) than the white variety from this sampling zone. Comparing the mean value of TFC between the three sampling zones, samples from the west Gojjam zone had a mean TFC-Q.E value higher than samples from East Gojjam and Awi zones.

Looking at the reported data by [25] for teff samples collected from USA and Bolivia, the TFC ranged between 62–67 mg rutin equivalent/100g in white teff and 106–116 mg rutin equivalent/100g) of brown teff samples. Similarly, a relatively higher concentration of total soluble TFC (41–63 mg C.E /100g of sample) was reported in white and brown varieties of teff collected from the Tigray region of Ethiopia [14].

The ANOVA test indicated in Table 2 confirmed significant variations ($p < 0.05$) in the total phenolic content among the 72 teff samples. The flavonoid content assessed by catechin equivalent (TFC-C.E) shows no significant difference ($p > 0.05$) between the samples taken from the three administrative zones. As indicated in Fig 5, the value of TPC and TFC-Q.E were found to be the highest in West Gojjam teff samples, followed by the Awi zone. TPC- G.A.E were highest in samples collected from the West Gojjam zone, followed by Awi and East Gojjam administrative zones. Looking at the error bars in Fig 5, the TPC, TFC-Q.E, and TFC-C.E in teff samples from the Awi zone and West Gojjam had wide variability within the sampling zones.

## Pearson correlation matrix of teff samples variables (72 samples × 3 variables)

The Pearson correlation matrices employing correlation coefficient (r) for the samples were utilized to correlate the influence of total phenol content on the total flavonoid content by

**Table 2. Analysis of variance (ANOVA) among 72 teff samples from three among three administrative zones.**

| | | ANOVA test | | | | |
|---|---|---|---|---|---|---|
| | | Sum of Squares | Degree of freedom | Mean Square | F- value | Significance test |
| TPC-GA | Between Groups | 10045 | 2 | 5022 | 30.38 | **0.00***  |
| | Within Groups | 11407 | 69 | 165 | | |
| | Total | 21451 | 71 | | | |
| TFC -CE | Between Groups | 216 | 2 | 108 | 2.54 | 0.08 |
| | Within Groups | 2928 | 69 | 42.43 | | |
| | Total | 3143 | 71 | | | |
| TFC-QE | Between Groups | 10333 | 2 | 5167 | 30.78 | **0.00***  |
| | Within Groups | 11581 | 69 | 168 | | |
| | Total | 21915 | 71 | | | |

*Significant at $p < 0.05$ by using ANOVA test.

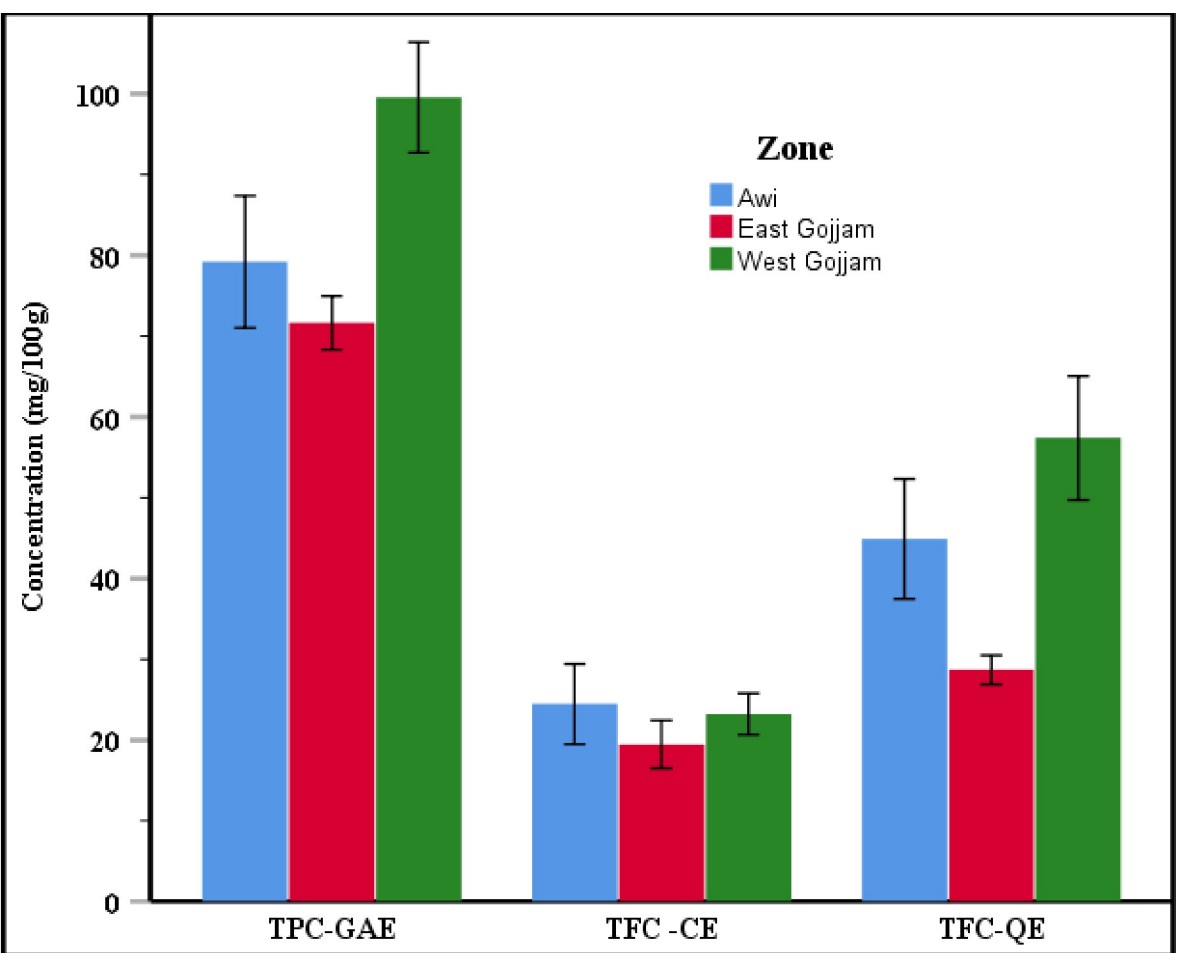

**Fig 5. Variation of phenolic contents with error bar graphs based on their administrative zones at a confidence interval of 95%.**

catechin and quercetin equivalents and measure the strength of a linear association between three variables, as shown in—Fig 6. The correlations between the total phenolic content and total flavonoid collected from the three administrative sampling areas of the teff samples were studied using linear regression correlation tests. The results are given in Fig 6. There was a positive correlation between total phenolic and total flavonoid content as catechin and quercetin equivalents. The variables are positively correlated and significant at (p = 0.01), as seen in the Pearson correlation matrix in, Fig 6, indicating that chemometric data treatment is possible. Among the various ways of multivariate analysis, hierarchical clustering can be applied to see the natural groupings of the samples and biplots to observe the relative positions of the variables and the distribution of the observations or data sets in a two-dimensional plot applied.

The 72 teff samples were trying to classify based on their geographical origin using hierarchical cluster analysis (HCA). In this classification model, the dendrogram was employed as a graphical tool to show the natural groupings or clusters based on the similarity ratios and Euclidean distances. As shown in Fig 7, the first and third clusters, encompassed by red and green rectangles, contain most of the samples obtained in the East Gojjam zone. Most teff samples from the West Gojjam zone were categorized on the last cluster or indicated by a violet color rectangle. The blue rectangle indicates the second cluster, containing teff samples from the Awi zone.

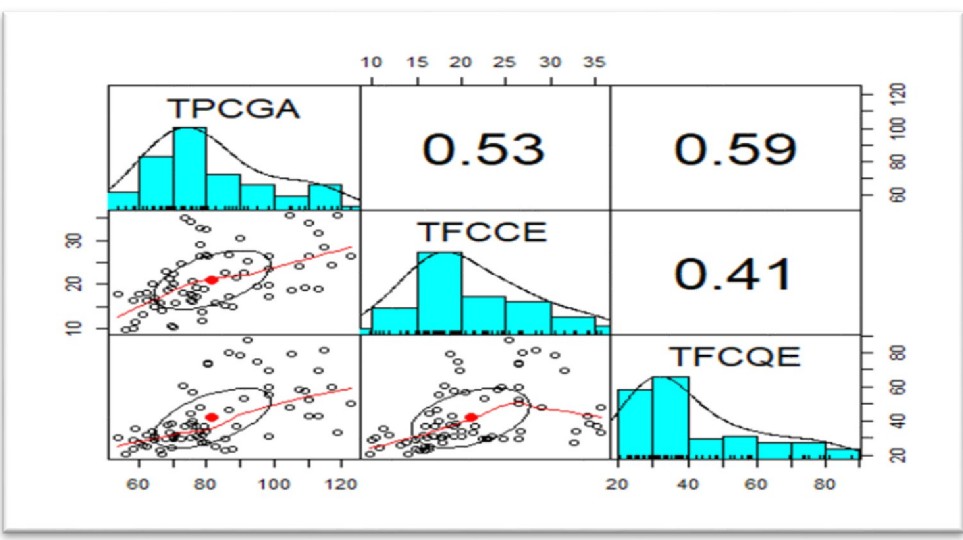

**Fig 6. Pearson correlation matrix of teff samples variables (72 samples × 3 variables).**

On the other hand, samples obtained from the Awi zone did not show a clear cluster. The lack of clear grouping of the teff samples of this zone might be because teff samples from this zone are of different varieties. In addition, there might be a difference in agroecological parameters with the sampling sub-districts.

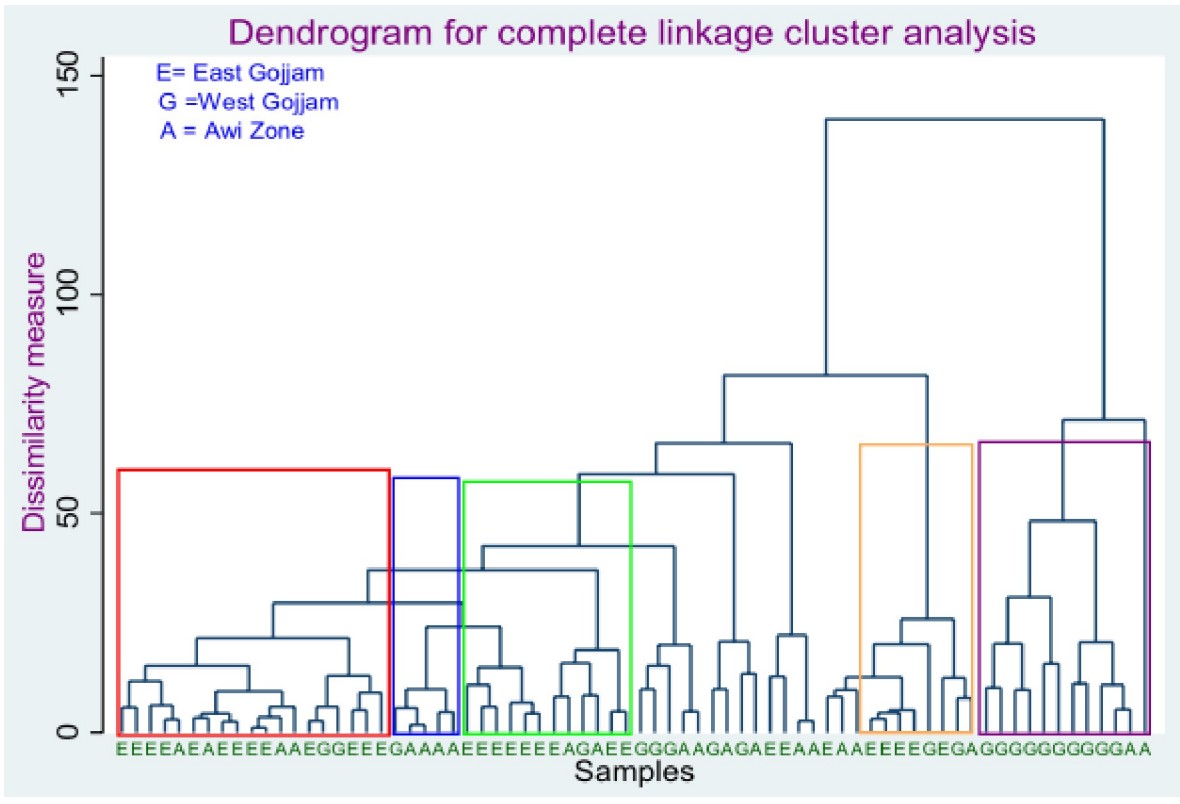

**Fig 7. Dendrogram from agglomerative hierarchical cluster analysis of 72 teff samples using phenolic parameters.**

Based on the biplots in Fig 8, the two components explain up to 87.2% of the variation in the three administrative zones. Thus, from Fig 8, principal component 1 explains 67.2% of the total variability of polyphenol content of the teff grains. Principal component 2 accounts for 20% of the variability of the teff grains' polyphenol content. The clusters with higher values of the principal component 1 are samples collected from West Gojjam and Awi zones, and principal component 2 is also associated with samples collected from the East Gojjam zone. Fig 8 shows a poor correlation between TFC-C.E and TFC-Q.E, while TPC demonstrates a higher correlation with TFC-Q.E. The distribution of the teff samples better cluster in a biplot based on their geographic origins. Most East Gojjam teff samples appeared between -2 and 1 in dimension1 (PC1) and slightly between -1 and 2 in dimension 2 (PC2). Fig 8 also demonstrates apparent clustering of the West and East Gojjam teff sample while samples collected from the Awi zone were intermixed with East Gojjam and West Gojjam zone. Most East Gojjam teff samples distribute at the left part, while those from West Gojjam are positioned at the right bottom side of the plot.

## Conclusion

The total phenolic and flavonoid content in 72 teff samples collected from three administrative zones were successfully investigated using UV-Vis spectroscopy. The teff samples (Kuncho variety or white in color) collected from the West Gojjam zone were rich in TPC-G.A.E and TFC-Q.E content. In contrast, the same variety of teff samples from the East Gojjam zone exhibited lower total phenolic and flavonoid content. The phenolic content in teff samples from the Awi zone was arranged in the order of Brown (mixed) teff > Red teff > Tseday teff (white variety). These differences might be attributed to variations in the genetic make-up of the samples, soil chemistry, and agro-ecology of the

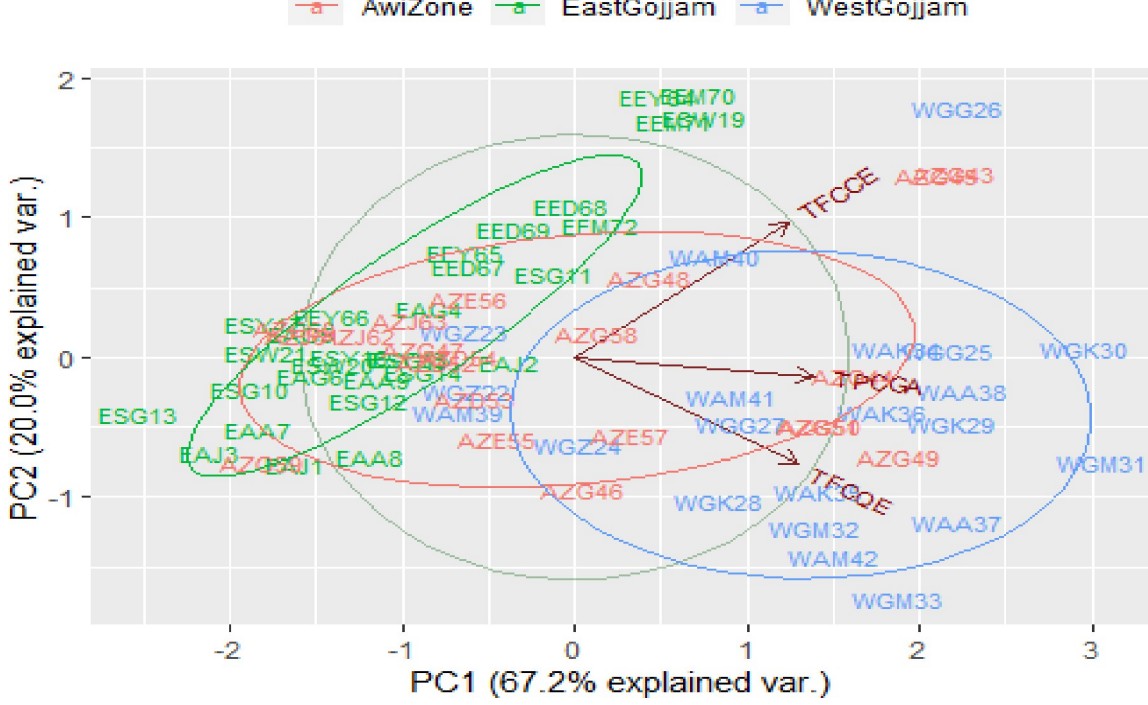

**Fig 8. Biplot of the phenolic observations and variables based on geographic origins.**

sampling zones. The result from the analysis of variance revealed that the flavonoid content determined by catechin equivalent (TFC-C.E) exhibits no significant difference ($p > 0.05$) between the samples obtained from the three administrative zones. However, total polyphenol and flavonoid contents by quercetin equivalent (TPC-Q.E) showed a significant difference at $p < 0.05$ among samples of the West Gojjam and East Gojjam zones. Thus, based on this study, teff samples collected from the West Gojjam zone had a high total polyphenol content and quercetin equivalent flavonoids.

## Acknowledgments

We would want to thank the Beauro of Agriculture of the Amhara Region, as well as its districts and sub-districts administrators, for giving information for this work.

## Author Contributions

**Conceptualization:** Minaleshewa Atlabachew, Tihitinna Asmellash.

**Data curation:** Chaltu Reta, Marie Yayinie.

**Formal analysis:** Chaltu Reta, Woldegiorgis Hilluf.

**Investigation:** Chaltu Reta, Woldegiorgis Hilluf.

**Methodology:** Chaltu Reta, Woldegiorgis Hilluf.

**Resources:** Tessera Alemneh Wubieneh.

**Software:** Marie Yayinie, Tessera Alemneh Wubieneh.

**Supervision:** Minaleshewa Atlabachew, Tihitinna Asmellash.

**Validation:** Tihitinna Asmellash, Marie Yayinie.

**Writing – original draft:** Chaltu Reta.

**Writing – review & editing:** Minaleshewa Atlabachew, Tihitinna Asmellash.

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
