## [Decision Letter · Decision Letter 0]

3 May 2022

PONE-D-22-08028Polyphenol and flavonoid content in major Teff [ Eragrostis tef (Zuccagni) Trotte r] varieties in Amhara Region, EthiopiaPLOS ONE

Dear Dr. Atlabachew,

Thank you for submitting your manuscript to PLOS ONE. After careful consideration, we feel that it has merit but does not fully meet PLOS ONE’s publication criteria as it currently stands. Therefore, we invite you to submit a revised version of the manuscript that addresses the points raised during the review process.

We look forward to receiving your revised manuscript.

Kind regards,

Patrizia Restani, Ph.D.

Academic Editor

PLOS ONE

Journal Requirements:

"The authors are thankful for the office for research and community service vice president of the Bahir Dar University for the financial support. In addition, Chaltu Reta is grateful to the college of the science of Bahir Dar University for sponsoring her PhD study."

 "Authors received no special fund from the university except providing laboratory facilities  and sampling costs."

5. We note that Figure 1 in your submission contain map/satellite images which may be copyrighted. All PLOS content is published under the Creative Commons Attribution License (CC BY 4.0), which means that the manuscript, images, and Supporting Information files will be freely available online, and any third party is permitted to access, download, copy, distribute, and use these materials in any way, even commercially, with proper attribution. For these reasons, we cannot publish previously copyrighted maps or satellite images created using proprietary data, such as Google software (Google Maps, Street View, and Earth). For more information, see our copyright guidelines: http://journals.plos.org/plosone/s/licenses-and-copyright.

a) You may seek permission from the original copyright holder of Figure 1 to publish the content specifically under the CC BY 4.0 license.  

6. Please include a separate caption for each figure in your manuscript.

Additional Editor Comments:

The paper needs careful revision. Authors are requested to respond to the individual comments of the reviewers. Please pay special attention to the statistical analysis

Reviewers' comments:

Reviewer's Responses to Questions

**Comments to the Author**

1. Is the manuscript technically sound, and do the data support the conclusions?

Reviewer #1: Partly

Reviewer #2: Partly

2. Has the statistical analysis been performed appropriately and rigorously? 

Reviewer #1: Yes

Reviewer #2: No

3. Have the authors made all data underlying the findings in their manuscript fully available?

Reviewer #1: Yes

Reviewer #2: Yes

4. Is the manuscript presented in an intelligible fashion and written in standard English?

Reviewer #1: Yes

Reviewer #2: Yes

5. Review Comments to the Author

Reviewer #1: The paper “Polyphenol and flavonoid content in major Teff [Eragrostis tef (Zuccagni) Trotter] varieties in Amhara Region, Ethiopia” provided interesting information about the polyphenol content of different Teff samples and varieties in relation to their region of provenience. For this aim a suitable number of samples was collected and analyzed.

I found the paper very interesting, however I have also identified some limitations that will be listed below:

- In the abstract the following sentence was reported “Such a difference is also evident in consumers' preference for teff grains from one location”. Report and explain better this consideration in “Results and discussions” or “Conclusions” paragraphs.

- In the introduction could be interesting to better explain the nutritional characteristics of Teef compared to other cereals: “Teff is gluten-free, abundant in minerals, vital amino acids, protein, and carbohydrates than other grains.”

- Explain the meaning of TTeff the first time that appeared

- Three different varieties of Teff were cited (and also analyzed). If the literature are reported differences, in term of phenolic compounds, between the Teff varieties (white, brown, and red) you could report them in the Introduction

- At the end of the introduction are cited, for the study, only two white Teff varieties

“Thus, this study was intended to quantify the level of total phenolic and total flavonoid contents in Kuncho teff (DZ-CR-387) and Tseday teff (DZ-CR-37) varieties collected from 21 sub-districts of three zones (East Gojjam, West Gojjam, and Awi)”

However, in Sample Collection and Pre-treatment paragraph were reported four different varieties:

“Four varieties of TTeff, such as Kuncho (white teff), Tseday (white teff), Brown (mixed), and Red teff were collected from model farmers in January 2020.”

Correct this difference and better explain the meaning of mixed for Brown Teff.

- In this study the determination of total flavonoid content was conducted expressing the results as catechin or quercetin equivalents.

In materials and methods was specify that some flavonoids such as quercetin, rutin, and galangin prefer to form a complex with aluminium chloride in an acid medium, therefore this class of flavonoids, was estimated in acidic conditions. However, in the description of method the used of acid medium was not reported. Pękal and Pyrzynska (2014) reported that acid or acetate solution could be used but, in some cases, only methanol or water was added. Correct or better explain the incongruity between the theory and the method applied.

Pękal A, Pyrzynska K. Evaluation of aluminium complexation reaction for flavonoid content assay. Food Anal Methods. 2014;7(9):1776-82.

- Some contradictions were observed between the text (in the results paragraph) and Tables 1 and 2; for example:

“On the other hand, the phenolic content recorded in samples from the West Gojjam and East Gojjam zones were ranged between 46.2 -92.2 mg/100 g and 45.95-128.5 mg/100 g, respectively.” The two zones were inverted and the data “45.95” was not reported in the Table 1.

“Comparing the mean value of TPC in samples from the three administrative zones, the highest average value of polyphenols was noted in the teff samples from the West Gojjam zone (95.04 mg G.A.E./100 g), followed by the Awi zone (79.37 mg G.A.E./100 g), and East Gojjam zone (71.39 mg G.A.E./100 g) (Table 2).” The data 79.37 and 71.39 don’t correspond to values reported in table 2

Carefully check all results.

- The sentence: “The lowest and highest mean concentrations of TFC were found in teff samples from East Gojjam (15.45 mg Q.E./100g) and the Awi zone (113.12 mg Q.E./100g), respectively, based on the mean value of total flavonoids at each sampling zone.” The results reported are not the mean concentrations listed in Table 2.

- Explain better how authors have conclude that:

“This implies that brown or mixed teff variety (from Awi zone) contained a higher total phenolic content than Kuncho teff variety (from East Gojjam zone) and Tseday teff variety (from West Gojjam zone). “(paragraph of Results and Discussion)

“the phenolic content in different teff samples was obtained in Brown teff > Kuncho teff > Tseday teff > Red Teff.” (paragraph of conclusions)

Overall, the manuscript is relatively well written however there are some typing and punctuation errors. For example, in Table 2, the Standard Error of the mean (indicated as SEM in the text), was reported as SE in the table and ESE in the legend.

Reviewer #2: Ref. No.: PONE-D-22-08028

Title: Polyphenol and flavonoid content in major Teff [Eragrostis tef (Zuccagni) Trotter]

varieties in Amhara Region, Ethiopia

The manuscript deals with a topic that could be of interest limited to the comparison of the polyphenol and flavonoid content in major Teff [Eragrostis tef (Zuccagni) Trotter] varieties in Amhara Region (Ethiopia).

The manuscript in some parts lacks clarity and an accurate revision is mandatory.

I have some questions and comments listed below:

Abstract:

the abstract is well organized.

In the line 3: ‘ ‘ region’s, please remove symbols.

Makeup: make-up

Last sentence of the abstract could be improved: “ These two dimensions (PC1 and PC2) explained the 87.2% of the total variance of the dataset”.

Introduction:

the introduction section is very rich but sometimes not clear and not presented in a logical way. Please revise this part.

“Vital” amino acids: please replace with essential.

“Its flour is mainly used to make "injera," Ethiopians' favourite national dish”: this has been said before, it is a repetition. Please modify in the manuscript.

“light to dark brown”: please modify “light brown to dark brown”.

“Reports indicate that total flavonoids are determined using two approaches: flavonols and flavones are selectively determined at 410–430 nm by reacting the sample solutions with AlCl3 in acidic or acetate solution. In contrast, flavan-3-ols (catechins and others), luteolin, and rutin could selectively react with AlCl3 solution in sodium nitrite and alkaline conditions, and measurement was done at 510 nm [24].

However, to the best of our knowledge, we did not find reports comparing the level of flavonoids determined by the two approaches simultaneously, either in teff or other products. Instead, the former method is most frequently reported to determine the total flavonoid content in various products”: this part should not be included in the introduction. Please move it to the discussion section.

“As information obtained from Amhara Region's Bureau of Agriculture, about 42 improved varieties of teff have been registered in the country, but only a few are widely cultivated in the region. Different types of teff are produced in all region zones”: In my opinion, this sentence is not so clear. Please better explain in the text.

Materials and Methods:

Please improve the resolution of Figure 1. In the map shown at the top left the names are not clearly legible. The sampling administration districts highlight in green in the maps are not visible.

In “Chemicals, Reagents, and Instruments”: ”Cary 60” Agilent, not Carry.

The Wetted sample: please remove the capital letter.

(porta centrifuge, Japan): please add the specifications of the centrifuge used in the previous paragraph.

and reported as mg gallic acid equivalent (mg G.A.E./100 g) of samples.

In “ Determination of total phenolic content by Folin–Ciocalteu assay”: please modify last sentence: “and reported as mg gallic acid equivalent for 100 g of samples (mg G.A.E./100 g).

Before reference [24], please remove the round bracket.

In “Determination of total flavonoid content as catechin equivalent“: “All measurements were performed in triplicate readings”. Were no analytical replicas of the spectrophotometric test made?

In “Determination of total flavonoid content as quercetin equivalent“: “All measurements were performed in triplicate readings”. Were no analytical replicas of the spectrophotometric test made?

Were quartz cuvettes used for the spectrophotometric readings? with which optical path? Please detail in the manuscript.

Statistical data analysis: Please express the acronyms TPC and TFC, TFC-QE first in the text.

Results and Discussion:

Figure 3: Concerning the calibration curve of catechin, the range of linearity was too short (abs from 0.00 to 0.2). Why were not considered standard solutions with higher concentration?

“The total phenolic content (TPC) is recorded as a milligram of gallic acid equivalent per 100 g of teff sample (mg of G.A.E./100 g teff). In addition, the catechin-like and quercetin-like total flavonoid contents are reported as milligram of catechin equivalent per 100 g of teff sample (mg of C.E./100 g) and milligram of quercetin equivalent per 100 g of teff sample (mg of Q.E./100 g teff) respectively”: this has already been said in materials and methods. Please remove it from Results and Discussion section.

Table1: please improve the formatting of table 1.

Figure 5 and Table 2 gives us the same information. Please remove Table 2 from the text (it could be added as supplementary materials). In Figure 5 please add error bars.

In the Discussion, Authors said that “The ANOVA test confirmed significant variations (P < 0.05) in its phenolic content among the 72 teff samples”. Where are reported the ANOVA results?

Table 3 is not clear. Please better explain in the text. Moreover, Principal component analysis has been applied but is not explain (in material and method section) and is not adequately discussed. This part needs to be improved.

Please check all the manuscript because there are several formatting errors. Please proofread the English, too.

6. PLOS authors have the option to publish the peer review history of their article (what does this mean?). If published, this will include your full peer review and any attached files.

Reviewer #1: No

Reviewer #2: No

---

## [Author Response · Author response to Decision Letter 0]

3 Jun 2022

We are very thankful to the reviewers and Editors for their appreciation of our manuscript entitled ' Polyphenol and flavonoid content in major Teff [Eragrostis tef (Zuccagni) Trotter]varieties in Amhara Region, Ethiopia, and the constructive criticism on certain issues. We are very grateful for your comments and feel that they added a lot of value and clarity to the overall organization of the manuscript. We agree with the reviewers and Editors that the comments raised should be addressed properly for better quality and acceptability of the manuscript. Therefore, we have revised our manuscript according to the reviewers' and editor’s comments, and the changes that have been made in the revised manuscript are track changed with different font colours.

The replies to the Reviewers' and editor’s comments are given carefully below:

Response to the Editor's comments

Comments #1. Please ensure that your manuscript meets PLOS ONE's style requirements, including those for file naming. 

Response # 1: We have checked and confirmed our manuscript against the journal's guideline

Comment # 2. We note that the grant information you provided in the 'Funding Information' and 'Financial Disclosure' sections do not match. 

When you resubmit, please ensure that you provide the correct grant numbers for the awards you received for your study in the 'Funding Information' section.

Thank you for stating the following in the Acknowledgments Section of your manuscript: 

"The authors are thankful for the office for research and community service vice president of the Bahir Dar University for the financial support. In addition, Chaltu Reta is grateful to the college of the science of Bahir Dar University for sponsoring her PhD study."

 "Authors received no special fund from the university except providing laboratory facilities and sampling costs."

Response # 2: We thank the Editor for raising this issue. We wish we could include the grant number in the funding statement. But we can't because we did not receive a formal fund from the university; rather, the research and community service vice president office of the university has covered costs related to sample collection, while the college of science provided the laboratory facilities because Chaltu Reta, a staff member of Bahir Dar University, is one of the students sponsored by the university. Thus, there was no grant number for this support and scholarship. Therefore, we have removed funds-related issues from this submission.

Comment #3: Please amend your list of authors on the manuscript to ensure that each author is linked to an affiliation. Authors' affiliations should reflect the institution where the work was done (if authors moved subsequently, you can also list the new affiliation stating "current affiliation:…." as necessary).

Response #3: Thank you so much for pointing out our mistake. We have now made the necessary corrections.

Comment # 4: We note that Figure 1 in your submission contain map/satellite images which may be copyrighted. All PLOS content is published under the Creative Commons Attribution License (CC BY 4.0), which means that the manuscript, images, and Supporting Information files will be freely available online, and any third party is permitted to access, download, copy, distribute, and use these materials in any way, even commercially, with proper attribution. For these reasons, we cannot publish previously copyrighted maps or satellite images created using proprietary data, such as Google software (Google Maps, Street View, and Earth). For more information, see our copyright guidelines: http://journals.plos.org/plosone/s/licenses-and-copyright.

Response # 4: We have now prepared our map using the GPS points obtained from the sampling districts and sub-districts. 

Comment #5: Please include a separate caption for each figure in your manuscript.

Response # 5: We have included a separate figure caption in the manuscript where the respective figure will be placed.

Comment # 6: Additional Editor Comments:

The paper needs careful revision. Authors are requested to respond to the individual comments of the reviewers. Please pay special attention to the statistical analysis

Response # 7: We have appreciated this comment. We have edited the manuscript considering the issues raised by the Editor and our respected reviewers. In addition, we have read the manuscript page by page and made substantial corrections.

Response to Reviewer: 1

Comment # 1: - In the abstract the following sentence was reported "Such a difference is also evident in consumers' preference for teff grains from one location". Report and explain better this consideration in "Results and discussions" or "Conclusions" paragraphs. 

Response #1: We thank the Reviewer for his/her comment. As per the Reviewer's suggestion, we have removed such a sentence from the abstract section. 

 Comment # 2: In the introduction could be interesting to better explain the nutritional characteristics of Teff compared to other cereals: "Teff is gluten-free, abundant in minerals, vital amino acids, protein, and carbohydrates than other grains."

Response #2: In the introduction section, we have included the following paragraph that compares the nutritional composition of teff as compared to other common cereals. 

Teff has a naturally higher nutritional value and is rich in essential amino acids and minerals such as Fe, Zn, Mg, and Ca compared to cereals such as maize and pearl millet. On the other hand, a comparable level of protein and carbohydrates is reported in teff grains compared to the most common cereal grains such as wheat, barley, sorghum, and pearl millet, and it is a gluten-free cereal grain [3, 5]. As many people are diagnosed with celiac disease and other forms of gluten sensitivity, the demand for gluten-free meals is increasing. As a result, the nutrient profile of teff grain suggests that it has a lot of potential for application in foods and beverages around the world [2, 6]. 

Comment # 3: Explain the meaning of TTeff the first time that appeared?

Response #3: Thank you for this comment. We have now defined the term "Teff" at the beginning (in the 1st lines of the abstract and introduction sections)

Comment # 4: Three different varieties of Teff were cited (and also analyzed). If the literature reported differences, in terms of phenolic compounds, between the Teff varieties (white, brown, and red) you could report them in the Introduction.

Response # 4: We have now included a paragraph showing the difference in nutritional composition between the three coloured varieties of teff 

"Different teff varieties are on the market. In terms of color, the most common teff varieties are white, brown and red. Brow teff is also known as mixed colored teff. It contains natural mixed white and red coloured teff grains. Report indicated that there is significant variation in minerlas and nutritional constituents between the three colored variants [ 14, 15]." 

Comment # 5: "Thus, this study was intended to quantify the level of total phenolic and total flavonoid contents in Kuncho teff (DZ-CR-387) and Tseday teff (DZ-CR-37) varieties collected from 21 sub-districts of three zones (East Gojjam, West Gojjam, and Awi). However, in Sample Collection and Pre-treatment paragraph were reported four different varieties: "Four varieties of TTeff, such as Kuncho (white teff), Tseday (white teff), Brown (mixed), and Red teff were collected from model farmers in January 2020." Correct this difference and better explain the meaning of mixed for Brown Teff.

Response # 5: The comments are accepted. We have now clarified the confusion. Such change can be depicted from the last page of the introduction section ( page 4) 

Comment # 6: In this study the determination of total flavonoid content was conducted expressing the results as catechin or quercetin equivalents. In materials and methods was specify that some flavonoids such as quercetin, rutin, and galangin prefer to form a complex with aluminium chloride in an acid medium, therefore this class of flavonoids, was estimated in acidic conditions. However, in the description of method the used of acid medium was not reported. Pękal and Pyrzynska (2014) reported that acid or acetate solution could be used but, in some cases, only methanol or water was added. Correct or better explain the incongruity between the theory and the method applied. Pękal A, Pyrzynska K. Evaluation of aluminum complexation reaction for flavonoid content assay. Food Anal Methods. 2014;7(9):1776-82.

Response #6: Thank you so much for pointing out the fault we made. We have clarified the confusion and edited the section dealing with flavonoid analysis.

Comment # 7: Some contradictions were observed between the text (in the results paragraph) and Tables 1 and 2; for example: "On the other hand, the phenolic content recorded in samples from the West Gojjam and East Gojjam zones were ranged between 46.2 -92.2 mg/100 g and 45.95-128.5 mg/100 g, respectively." The two zones were inverted and the data "45.95" was not reported in Table 1.

Response#7: We appreciate our esteemed Reviewer's insightful view. The issues raised are fully noted, and we have corrected them in the reviewed manuscript.

Comment # 8. Explain better how authors have conclude that: "This implies that brown or mixed teff variety (from Awi zone) contained a higher total phenolic content than Kuncho teff variety (from East Gojjam zone) and Tseday teff variety (from West Gojjam zone). "(paragraph of Results and Discussion) "the phenolic content in different teff samples was obtained in Brown teff > Kuncho teff > Tseday teff > Red Teff." (paragraph of conclusions)

Response #8: comments were accepted and incorporated into the revised manuscript. We have clarified those confusions in the abstract section and the result and discussion section.

Comment # 9: Overall, the manuscript is relatively well written however there are some typing and punctuation errors. For example, in Table 2, the Standard Error of the mean (indicated as SEM in the text), was reported as SE in the table and ESE in the legend.

Response # 9: Thank you for your wonderful comments. We have read the manuscript page by page and significantly improved the typographic and spelling errors. 

Reviewer 2.

Abstract:

Comment #1: In the line 3: '‘ region's, please remove symbols.

Response #1: We are very grateful to the Reviewer for pointing out this comment. The symbols have now been removed.

Comment #2: Makeup: make-up

Response #2: Thank you so much. We have converted the word “ Makeup” to “make-up”

Comment # 3: The Last sentence of the abstract could be improved: "These two dimensions (PC1 and PC2) explained the 87.2% of the total variance of the dataset".

Response #3 We thank the reviewers for pointing out this suggestion. We have now removed the stated sentence.

Introduction 

Comment # 4: The introduction section is very rich but sometimes not clear and not presented in a logical way. Please revise this part

Response #4 We thank the Reviewer comment for his/her comment. We have thoroughly edited the introduction section and other sections of the manuscript.

Comment #5: "Vital" amino acids: please replace them with essential.

Response #5: We have replaced the phrase “ Vital amino acids” with “essential amino acids”.

Comment # 6: "Its flour is mainly used to make "injera," Ethiopians' favorite national dish": this has been said before, it is a repetition. Please modify the manuscript.

Response #6: We thank our Review for this comment. We have deleted this repeated sentence.

Comment #7: "light to dark brown": please modify "light brown to dark brown".

Response #7: We have accepted the comment and corrected it accordingly. 

Comment # 8. Reports indicate that total flavonoids are determined using two approaches: flavonols and flavones are selectively determined at 410–430 nm by reacting the sample solutions with AlCl3 in acidic or acetate solution. In contrast, flavan-3-ols (catechins and others), luteolin, and rutin could selectively react with AlCl3 solution in sodium nitrite and alkaline conditions, and measurement was done at 510 nm [24].

However, to the best of our knowledge, we did not find reports comparing the level of flavonoids determined by the two approaches simultaneously, either in teff or other products. Instead, the former method is most frequently reported to determine the total flavonoid content in various products": this part should not be included in the introduction. Please move it to the discussion section.

Response #8: We thank our esteemed Reviewer for the suggestion. We took the following paragraph to the result and discussion section.”

“ In this study, the total flavonoids in the 72 teff samples were determined following two approaches. Reports indicate that not all flavonoids react with AlCl3 in the presence of sodium nitrite and sodium hydroxide. Instead, flavonols and flavones classes of flavonoids such as quercetin, quercetrin, galangin, and rutin, are selectively determined at 410–430 nm by reacting the sample solutions with AlCl3 in acidic, acetate buffer, or methanolic solutions. In contrast, flavan-3-ols (catechins and others) could selectively react with AlCl3 solution in sodium nitrite and alkaline conditions, and measurement can be done at 510 nm[27]. Thus, flavonoid data obtained with only one of the approaches could not reflect the total falvonoids content in a given sample”.

Comment # 9: "As information obtained from Amhara Region's Bureau of Agriculture, about 42 improved varieties of teff have been registered in the country, but only a few are widely cultivated in the region. Different types of teff are produced in all region zones": In my opinion, this sentence is not so clear. Please better explain in the text.

Response # 9: We appreciate the Reviewer’s comment and concerns. We have now clarified the stated paragraph as follows: 

“As information obtained from Amhara Region's Bureau of Agriculture, about 42 improved varieties of teff have been registered in the country, but only a few are widely cultivated in the region.. Among the varieties of teff, Kuncho teff (DZ-CR-387) and Tseday teff (DZ-CR-37) are white by color and widely cultivated in the northwestern part of the Amhara region. Kuncho teff is cultivated in East Gojjam and West Gojjam zones, while Tseday teff variety is mainly grown in the Awi zone. In addition, red teff and brown teff varieties are also cultivated in the region, but consumers less prefer these colored varieties. Brown (mixed) teff is a variety of teff that is locally called “sergegna” and naturally exists asa mixture of white and red teff varieties. Kuncho and Tseday teff varieties have a high market price and are most preferred by consumers. But the same variety of teff produced in different zones and districts have different qualitiesand market prices, hence having consumer preferences. Since the Amhara region is one of the most teff-producing areas of the country, there is no report on the level of total phenolics and flavonoids in different teff varieties cultivated in this region”

Materials and Methods:

Comment #10: Please improve the resolution of Figure 1. In the map shown at the top left the names are not clearly legible. The sampling administration districts highlight in green in the maps are not visible.

Response #10: Thank you for the comment. Figure 1 has been replaced with a well-resolved figure. 

Comment #11: In "Chemicals, Reagents, and Instruments":" Cary 60" Agilent, not Carry.

Response #11: It is corrected accordingly. Thank you once again.

Comment #12: The Wetted sample: please remove the capital letter.

Response # 12: It is corrected accordingly. Thank you for the comment

Comment #13: (porta centrifuge, Japan): please add the specifications of the centrifuge used in the previous paragraph.

Response # 13: A specification for the centrifuge is added in the materials and method section.

Comment #14: In "Determination of total phenolic content by Folin–Ciocalteu assay": please modify last sentence: "and reported as mg gallic acid equivalent for 100 g of samples (mg G.A.E./100 g).

Response #14: It is corrected accordingly.

Comment #15: Before reference [24], please remove the round bracket.

Response #15: Thank you, the round bracket before [24], is removed. 

Comment #16: In "Determination of total flavonoid content as catechin equivalent ": "All measurements were performed in triplicate readings". Were no analytical replicas of the spectrophotometric test made?

Response #16: We thank our esteemed Reviewer. The sentence was confusing. We have now clarified it as follows in the revised manuscript.

“Each sample was analysed in triplicate.”

Comment #17: In "Determination of total flavonoid content as quercetin equivalent ": "All measurements were performed in triplicate readings". Were no analytical replicas of the spectrophotometric test made?

Response #17: thank you for your comment. We have corrected it as follows:

 “Each sample analysed in triplicate”

Comment #18: Were quartz cuvettes used for the spectrophotometric readings? with which optical path? Please detail in the manuscript.

Response #18: Thank you for pointing out this comment. A description about the sample holder is given under the “ Chemicals, reagent and instrument section” as follows:

“. Quartz cuvette with 1 cm path length was as sample holder”.

Comment #19: Statistical data analysis: Please express the acronyms TPC and TFC, TFC-QE first in the text. 

Response #19: Acronyms of TPC, TFC-Q.E and TFC-C.E have now been defined in the method section and statistical data analysis section. 

Comment #20: Figure 3: Concerning the calibration curve of catechin, the range of linearity was too short (abs from 0.00 to 0.2). Why were not considered standard solutions with higher concentrations?

Response: We would like to appreciate our Reviewer’s comment. Before the calibration curve range was fixed, an initial screening test was performed for some samples. Then it was noted that the samples absorbance values were lower than 0.2. Then we decided to take several concertation of catechin standards that provide an absorbance value between 0.00 and 0.20. Otherwise, the linear range might go beyond this range.

Comment #21: "The total phenolic content (TPC) is recorded as a milligram of gallic acid equivalent per 100 g of teff sample (mg of G.A.E./100 g teff). In addition, the catechin-like and quercetin-like total flavonoid contents are reported as milligram of catechin equivalent per 100 g of teff sample (mg of C.E./100 g) and milligram of quercetin equivalent per 100 g of teff sample (mg of Q.E./100 g teff) respectively": this has already been said in materials and methods. Please remove it from the Results and Discussion section.

Response# 21 Thank you so much. The redundant sentence has been deleted from the revised manuscript. 

Comment # 22. Table1: please improve the formatting of table 1.

Response# 22: We have appreciated the comment. We have now improved the formatting of table 1.

Comment # 23. Figure 5 and Table 2 gives us the same information. Please remove Table 2 from the text (it could be added as supplementary materials). In Figure 5 please add error bars.

Response# 23: Table 2 was removed, and the error bar was added to the graph in the revised manuscript.

Comment # 24: In the Discussion, Authors said that "The ANOVA test confirmed significant variations (P < 0.05) in its phenolic content among the 72 teff samples". Where are reported the ANOVA results?

Response# 24:Thank you for your comments, and the ANOVA results have now been incorporated into the revised manuscripts ( page 16).

Comment # 25. Table 3 is not clear. Please better explain in the text. Moreover, Principal component analysis has been applied but is not explained (in the material and method section) and is not adequately discussed. This part needs to be improved.

Response# 25: We are thankful to our respected Reviewer for raising this issue. We have significantly improved the discussion part of table 3, and the PCA. We have also explained the PCA in the method section.

Comment # 26. Please check all the manuscript because there are several formatting errors. Please proofread the English, too.

Response# 26: We have gone through the manuscript line by line and significantly revised the manuscript.

---

## [Decision Letter · Decision Letter 1]

30 Jun 2022

PONE-D-22-08028R1Polyphenol and flavonoid content in major Teff [ Eragrostis tef (Zuccagni) Trotte r] varieties in Amhara Region, EthiopiaPLOS ONE

Dear Dr. Minaleshewa Atlabachew

Thank you for submitting your manuscript to PLOS ONE. After careful consideration, we feel that it has merit but does not fully meet PLOS ONE’s publication criteria as it currently stands. Therefore, we invite you to submit a revised version of the manuscript that addresses the points raised during the review process.

We look forward to receiving your revised manuscript.

Kind regards,

Patrizia Restani, Ph.D.

Academic Editor

PLOS ONE

Journal Requirements:

Additional Editor Comments (if provided):

The paper has improved and now requires only minor changes of the text.

Reviewers' comments:

Reviewer's Responses to Questions

**Comments to the Author**

1. If the authors have adequately addressed your comments raised in a previous round of review and you feel that this manuscript is now acceptable for publication, you may indicate that here to bypass the “Comments to the Author” section, enter your conflict of interest statement in the “Confidential to Editor” section, and submit your "Accept" recommendation.

Reviewer #1: (No Response)

Reviewer #2: All comments have been addressed

2. Is the manuscript technically sound, and do the data support the conclusions?

Reviewer #1: Yes

Reviewer #2: Yes

3. Has the statistical analysis been performed appropriately and rigorously? 

Reviewer #1: Yes

Reviewer #2: Yes

4. Have the authors made all data underlying the findings in their manuscript fully available?

Reviewer #1: Yes

Reviewer #2: Yes

5. Is the manuscript presented in an intelligible fashion and written in standard English?

Reviewer #1: Yes

Reviewer #2: Yes

6. Review Comments to the Author

Reviewer #1: The paper “Polyphenol and flavonoid content in major Teff [Eragrostis tef (Zuccagni) Trotter] varieties in Amhara Region, Ethiopia” was improve by authors and all the suggestions were amended.

The paper is very interesting; however, some little corrections could be done:

- some little typing and punctuation errors were still observed (e.g. in the tile and in the test be lacking spaces);

- In the Introduction replace “It's a chasmogamous C4 annual cereal with a fibrous…” with “Teff is a chasmogamous C4 annual…”;

- In the Results and Discussion, the Figure 1 on gallic acid results is the Figure 2;

- In the Results and Discussion, I suggest to modify the following sentence “This implies that brown or mixed teff variety (from Awi zone) contained a higher total phenolic content than Kuncho teff variety (from East Gojjam and West Gojjam zone) and Tseday teff variety (from Awi zone).” Some samples from West Gojjam zone have an high TPC, indeed the mean value of TPC and TFC-Q.E were found to be the highest in West Gojjam teff samples compared to the samples from other zones.

Among varieties from Awi zone, the brown samples showed the highest TPC. Similar comments could be also arranged for the TFC.

- In the Results and Discussion, in the sentence “On the other hand, relatively comparable TPC values with the present study were reported by [14] for white and brown teff varieties collected from the Tigray region of Ethiopia.” could be interest to specify the values reported by the reference authors.

- Table 3 must be insert in the text as a Figure or as a Table?

Reviewer #2: Authors responded to comments by significantly improving the quality of the manuscript. In my opinion, the work in this formcan be accepted for its publication in Plos One.

7. PLOS authors have the option to publish the peer review history of their article (what does this mean?). If published, this will include your full peer review and any attached files.

Reviewer #1: No

Reviewer #2: No

---

## [Author Response · Author response to Decision Letter 1]

5 Jul 2022

Replies to the technical comments of the reviewers and Editors

Ref. No.: PONE-D-22-08028

Title: Polyphenol and flavonoid content in major Teff [Eragrostis tef (Zuccagni) Trotter]

varieties in Amhara Region, Ethiopia

We are very thankful to the reviewers and Editors for their appreciation of our manuscript entitled Polyphenol and flavonoid content in major Teff [Eragrostis tef (Zuccagni) Trotter]varieties in Amhara Region, Ethiopia, and the constructive criticism on certain issues. The replies to the Reviewers' and editor's comments are given carefully below:

Response to reviewer #1 comments.

Comments #1: Some little typing and punctuation errors were still observed (e.g. in the tile and in the test lacking spaces)

Response #1: Significant typographic errors and punctuation corrections have been taken throughout the manuscript.

Comments #2: In the Introduction replace "It's a chasmogamous C4 annual cereal with a fibrous…" with "Teff is a chasmogamous C4 annual…";

Response #2: The reviewer's suggestion is incorporated in the revised manuscript.

Comment #3: In the Results and Discussion, Figure 1 on gallic acid results is Figure 2;

 Response # 3: Figure 1 refers to a map of the sampling areas, while Figure 2 refers to the gallic acid standard's calibration curve and UV spectra. 

Comment #4: In the Results and Discussion, I suggest to modify the following sentence "This implies that brown or mixed teff variety (from Awi zone) contained a higher total phenolic content than Kuncho teff variety (from East Gojjam and West Gojjam zone) and Tseday teff variety (from Awi zone)." Some samples from the West Gojjam zone have a high TPC, indeed the mean value of TPC and TFC-Q.E were found to be the highest in West Gojjam teff samples compared to the samples from other zones

Response #4: The brown or mixed teff variety from Awi zone contained a higher total phenolic content than the white teff (Tseday variety ) collected from the Awi zone. The mean TPC values were compared between the two white teff varieties (Kuncho and Tseday) collected from the three administrative zones. It was noted that the Kuncho teff variety collected from the West Gojjam Zone contained a higher mean TPC value than the same Kuncho variety collected from the East Gojjam zone and the other variety (Tsedey) from the Awi zone. 

Comment #5: Among varieties from the Awi zone, the brown samples showed the highest TPC. Similar comments could be also arranged for the TFC.

Response #5: Thank you once again for pointing out this. We have modified the phrase as follow: Similar to the TPC value, the brown (mixed) teff samples from the Awi zone had higher flavonoid contents (Table 1) than the white variety from this sampling zone. Comparing the mean value of TFC between the three sampling zones, samples from the west Gojjam zone had a mean TFC-Q.E value higher than samples from East Gojjam and Awi zones

Comment #6: In the Results and Discussion, in the sentence "On the other hand, relatively comparable TPC values with the present study were reported by [14] for white and brown teff varieties collected from the Tigray region of Ethiopia." could be interest to specify the values reported by the reference authors

Response #6: Thank you for your comments. We have now incorporated the data reported by [14} as follows. On the other hand, 263 mg GAE/100g and 448 mg GAE /100g, respectively, for white (Tseday type) and brown teff varieties collected from the Tigray region of Ethiopia were reported by [14].

 Comment #7: Table 3 must be insert in the text as a Figure or as a Table?

Response #7: We are grateful for this comment. We have changed table 3 into Figure 6. Then the previous figure 6 is labeled as figure 7, and the last figure 7 is captioned figure 8 in the revised manuscript. This change has been corrected in the text too.

---

## [Editor Report · Decision Letter 2]

12 Jul 2022

Polyphenol and flavonoid content in major Teff [ Eragrostis tef (Zuccagni) Trotte r] varieties in Amhara Region, Ethiopia

PONE-D-22-08028R2

Dear Dr. Atlabachew

We’re pleased to inform you that your manuscript has been judged scientifically suitable for publication and will be formally accepted for publication once it meets all outstanding technical requirements.

Kind regards,

Patrizia Restani, Ph.D.

Academic Editor

PLOS ONE
---

## [Editor Report · Acceptance letter]

18 Jul 2022

PONE-D-22-08028R2 

Polyphenol and flavonoid content in major Teff [*Eragrostis tef (Zuccagni)* Trotter] varieties in Amhara Region, Ethiopia 

Dear Dr. Atlabachew:

I'm pleased to inform you that your manuscript has been deemed suitable for publication in PLOS ONE. Congratulations! Your manuscript is now with our production department. 

Kind regards, 

on behalf of

Professor Patrizia Restani 

Academic Editor

PLOS ONE